# Associations of Perceived Socially Unfavorable Attitudes toward Homosexuality and Same-Sex Marriage with Suicidal Ideation in Taiwanese People before and after Same-Sex Marriage Referendums

**DOI:** 10.3390/ijerph17031047

**Published:** 2020-02-07

**Authors:** Nai-Ying Ko, I-Hsuan Lin, Yu-Te Huang, Mu-Hong Chen, Wei-Hsin Lu, Cheng-Fang Yen

**Affiliations:** 1Departments of Nursing, College of Medicine, National Cheng Kung University and Hospital, Tainan 70101, Taiwan; nyko@mail.ncku.edu.tw; 2Center of Infection Control, National Cheng Kung University Hospital, Tainan 70101, Taiwan; 3Department of Psychiatry, Yuan’s General Hospital, Kaohsiung 80249, Taiwan; ihreneelin@gmail.com; 4Department of Health Business Administration, Meiho University, Pingtung 91202, Taiwan; 5Department of Social Work and Social Administration, The University of Hong Kong, Hong Kong RM543, China; Yuhuang@hku.hk; 6Department of Psychiatry, Taipei Veterans General Hospital, Taipei 11217, Taiwan; kremer7119@gmail.com; 7Division of Psychiatry, School of Medicine, National Yang-Ming University, Taipei 11221, Taiwan; 8Department of Psychiatry, Ditmanson Medical Foundation Chia-Yi Christian Hospital, Chia-Yi City 60002, Taiwan; 9Department of Senior Citizen Service Management, Chia Nan University of Pharmacy and Science, Tainan 71710, Taiwan; 10Department of Psychiatry, School of Medicine, College of Medicine, Kaohsiung Medical University, Kaohsiung 80708, Taiwan; 11Department of Psychiatry, Kaohsiung Medical University Hospital, Kaohsiung 80708, Taiwan

**Keywords:** suicidal ideation, homosexuality, same-sex marriage, sexual orientation

## Abstract

This study examined the associations of perceived socially unfavorable attitudes toward homosexuality and same-sex marriage with suicidal ideation in non-heterosexual and heterosexual participants from first (Wave 1, 23 months prior to same-sex marriage referendums) and second (Wave 2, one week after the referendums) wave surveys in Taiwan. Data provided by 3239 participants in Wave 1 and 1337 participants in Wave 2 who were recruited through a Facebook advertisement were analyzed. Participants completed an online questionnaire assessing suicidal ideation and perceived unfavorable attitudes toward homosexuality and same-sex marriage from Taiwanese society, heterosexual friends, and family members. The results indicate that perceived unfavorable attitudes toward homosexuality from Taiwanese society, heterosexual friends, and family members were positively associated with suicidal ideation among non-heterosexual individuals in the first but not the second survey. In addition, among non-heterosexual individuals, such attitudes toward same-sex marriage in family members and in heterosexual friends were positively associated with suicidal ideation in the Wave 1 and Wave 2 surveys, respectively. Perceived unfavorable attitudes toward homosexuality and same-sex marriage in heterosexual friends were associated with suicidal ideation in heterosexual participants with a favorable attitude but not in those individuals with an unfavorable attitude toward homosexuality, in both surveys. Perceived socially unfavorable attitudes toward homosexuality and same-sex marriage were significantly associated with suicidal ideation before and after same-sex marriage referendums; however, the associations varied between non-heterosexual and heterosexual individuals.

## 1. Introduction

### 1.1. Perceived Unfavorable Attitudes toward Homosexuality and Suicidal Ideation in Lesbian, Gay, and Bisexual Individuals

High rates of suicidal ideation and attempts among lesbian, gay, and bisexual (LGB) individuals have been reported in both Western [1,2] and Asian [3,4] societies. Meta-analysis has revealed that LGB individuals report significantly higher rates of suicidal ideation and attempts than their heterosexual counterparts [5,6] and the general population [7]. According to minority stress theory [8], perceived socially negative attitudes and experiences of discrimination are the main causes of increased suicidal risk in LGB individuals. Perceived stigma [9], expected stigma [10], and internalized stigma [11] contribute to this increase of suicidal risk. 

The identification of risk factors for suicidality is essential for developing prevention and intervention strategies in order to reduce suicidality in LGB individuals. According to ecological systems theory [12], suicidality can result from complex interactions between LGB individuals and their environments. LGB individuals perceive unfavorable attitudes toward homosexuality from multiple sources, such as society, friends, and family members. Whether the source of perceived unfavorable attitudes influences suicidal ideation in LGB individuals warrants study.

### 1.2. Perceived Unfavorable Attitudes toward Same-Sex Marriage and Suicidal Ideation in LGB Individuals

Same-sex marriage bans are one type of structural-level discrimination that deny LGB individuals legal, financial, health, and other benefits associated with marriage [13,14,15]. Research in the United States has found that same-sex marriage bans were associated with increased rates of psychiatric disorders [16]. Research on men who have sex with men (MSM) in China also reported that the illegal status of same-sex marriage was one cause of suicidal behaviors in MSM [3]. Social and legal recognition of same-sex relationships can reduce discrimination against LGB individuals [17] and significantly decrease the use of and expenditure on mental health care among MSM [18]. The results of these previous studies indicate the positive role of same-sex marriage for health in LGB individuals.

Whether perceived socially unfavorable attitudes toward same-sex marriage are associated with suicidal ideation in LGB individuals warrants further study. Moreover, whether perceived unfavorable attitudes toward same-sex marriage from various sources show different associations with suicidal ideation in LGB individuals also warrants further investigation.

### 1.3. Does it Matter whether Heterosexual Individuals Perceive Socially Unfavorable Attitudes toward a Sexual Minority?

Few studies have examined the association between an anti-LGB climate and mental health in heterosexual individuals. One USA study found that LGB, but not heterosexual, individuals experience significantly increased mental health problems after the passing of constitutional amendments banning same-sex marriage [16]. However, another USA study revealed that the legalization of same-sex marriage was associated with a 7% relative reduction of suicide attempts in both heterosexual and homosexual high school students [19]. It is reasonable to hypothesize that heterosexual individuals who report a friendly attitude toward sexual minorities believe that unfavorable attitudes toward homosexuality and same-sex marriage violate social justice, and therefore they are upset by them. This association between perceived socially unfavorable attitude towards same-sex marriage and suicidal ideation warrants further study.

### 1.4. Same-Sex Marriage Campaign and Referendums in Taiwan

Sexual minority rights campaigners, in Taiwan, have strived for same-sex marriage legalization since the end of the 1980s. In May 2017, the Justices of the Constitutional Court in Taiwan determined that the Civil Code that bars same-sex marriage violates the right to equality, and therefore is unconstitutional. It ruled that same-sex marriage be legalized within two years. In response to the decision, a group against same-sex marriage, mainly composed of Christians, soon drafted two referendums arguing that marriage as defined in the Civil Code should be restricted to a union between one man and one woman and that legal reform involving same-sex partnerships should be made without changing the Civil Code. By contrast, a group lobbying for marriage equality drafted a referendum arguing that separate legislation for same-sex partnerships would be discriminatory. The results of the 24 November, 2018 vote suggested that the two referendums against same-sex marriage had considerably stronger support from people in Taiwan than the one in favor of marriage equality.

Although the results of the referendums discouraged LGB rights campaigners, the referendums attracted people’s attention and inspired nationwide debates on homosexuality and same-sex marriage. For most people in Taiwan, it was the first time that they had paid close attention to and expressed attitudes toward homosexuality and same-sex marriage. The same-sex marriage referendums provide an opportunity to examine whether perceived unfavorable attitudes toward homosexuality and same-sex marriage from Taiwanese society, heterosexual friends, and family members are significantly associated with suicidal ideation among people in Taiwan.

### 1.5. Aims and Hypotheses of the Study

This study used data from the Investigation on the Attitude Toward Same-Sex Marriage in Taiwan, which was a two-wave online survey of attitudes toward homosexuality, same-sex marriage, and mental health in Taiwan [20]. The first wave (Wave 1) was conducted from 1 to 31 January, 2017, 23 months before the referendums. The second wave (Wave 2) was conducted from 1 to 31 December, 2018, one week after the referendums. This study examined the associations of perceived socially unfavorable attitudes toward homosexuality and same-sex marriage with suicidal ideation in non-heterosexual and heterosexual participants in the two surveys. We tested four hypotheses. First, on the basis of the reviewed literature, we hypothesize that perceived unfavorable attitudes toward homosexuality and same-sex marriage, from Taiwanese society, heterosexual friends, and family members is positively associated with suicidal ideation in non-heterosexual and heterosexual individuals in both the Wave 1 and Wave 2 surveys. Second, given that individuals are differentially affected by the source of attitudes, we hypothesize that the source of these perceived attitudes influences suicidal ideation. Third, given also that non-heterosexual and heterosexual individuals show differing vulnerabilities to social attitudes toward sexual minorities, we hypothesize that the associations of such attitudes with suicidal ideation vary between such individuals. Fourth, because of keen debates and anti-LGB propaganda during the referendums, we hypothesize that the association of these attitudes with suicidal ideation is different before and after the same-sex marriage referendums.

## 2. Methods

### 2.1. Participants and Procedure

The method used to recruit participants is described elsewhere [20]. In brief, participants aged at least 20 years were recruited into the two-wave online survey through a Facebook advertisement. The Facebook advertisement included a headline, main text, pop-up banner, and weblink to the questionnaire website. The advertisement appeared in the News Feed of Facebook, which is a streaming list of updates from the users’ connections and advertisers. News feed advertisements are effective recruitment metrics for studies [21]. We targeted the advertisement to Facebook users by location (Taiwan) and language (Chinese). The deduplication protocol used to identify multiple submissions and preserve data integrity included the cross-validation of the eligibility of key variables, examination of discrepancies in key data, and checking for unusually fast completion times (<10 min) [22]. Moreover, an Internet Protocol address was registered, and the questionnaires could be completed only once from that address.

Participants were not given any incentives for participation. The study was approved by the Institutional Review Board (IRB) of Kaohsiung Medical University Hospital. The study design allowed respondents to respond to the recruitment advertisement and questionnaire anonymously, and their personal information was kept secure. Owing to the anonymity of the participants, we could not determine how many participants responded to both surveys. Therefore, the data of the two survey waves were analyzed independently. The IRB, thus, agreed that obtaining informed consent from the respondents was unnecessary.

### 2.2. Measures

#### 2.2.1. Suicidal Ideation

We used the question “Do you have any suicide ideation?” from the Revised 5-item Brief Symptom Rating Scale to inquire after participants’ suicidal ideation during the previous week. Participants were asked to rate the severity of suicidal ideation on a 5-point scale: 0, *none at all; 1, a little bit; 2, moderately; 3, quite a bit; and 4, extremely* [23]. Participants who rated the item ≥2 were classified as experiencing significant suicidal ideation. The Institutional Review Boards (IRBs) of Kaohsiung Medical University (KMUHIRB-EXEMPT(II)-20160065) approved the study.

#### 2.2.2. Unfavorable Attitudes toward Homosexuality and Same-Sex Marriage

We used six questions to determine how participants perceived the attitudes of Taiwanese society, their heterosexual friends, and their family members toward homosexuality (“To what degree does Taiwanese society accept homosexuality?”, “To what degree do your heterosexual friends accept homosexuality?”, and “To what degree do your family members accept homosexuality?”) and same-sex marriage (“To what degree does Taiwanese society accept same-sex marriage?”, “To what degree do your heterosexual friends accept same-sex marriage?”, and “To what degree do your family members accept same-sex marriage?”). Participants indicated acceptance of homosexuality and same-sex marriage on a 5-point Likert scale ranging from 0 (very low) to 4 (very high). The responses to the six questions for social attitudes toward homosexuality and same-sex marriage were significantly skewed. For the purposes of this study, participants who scored 0 to 2 and who scored 3 or 4 were classified into the group who perceived unfavorable and favorable attitudes toward homosexuality and same-sex marriage from Taiwanese society, heterosexual friends, and family members, respectively. We also asked heterosexual participants to rate their attitude toward homosexuality using the same scale.

#### 2.2.3. Demographic Variables

Data on participants’ gender, age, and sexual orientation (heterosexual, bisexual, homosexual, pansexual, asexual, and questioning) were collected. For sexual orientation, participants were classified into non-heterosexual (including bisexual, homosexual, and others) and heterosexual groups.

### 2.3. Statistical Analysis

Data analysis was performed using SPSS Version 20.0 (SPSS Inc., Chicago, IL, USA). Gender, age, suicidal ideation, and perceived unfavorable attitudes toward homosexuality and same-sex marriage in the Wave 1 and Wave 2 surveys were compared using Chi-square and *t* tests. The associations of the perceived attitudes from Taiwanese society, heterosexual friends, and family members with suicidal ideation in non-heterosexual and heterosexual participants before (Wave 1 survey) and after (Wave 2 survey) the same-sex marriage referendums were examined using logistic regression analysis. A *p* value of 0.05 or lower was used to indicate significance. Odds ratios (ORs) and 95% confidence intervals (CIs) were used to describe the results of the analysis.

## 3. Results

### 3.1. Rates of Suicidal Ideation and Perceived Unfavorable Attitudes toward Homosexuality and Same-Sex Marriage

The data for 3239 participants (1443 heterosexual and 1796 non-heterosexual individuals) in Wave 1 surveys and 1337 participants (539 heterosexual and 798 non-heterosexual individuals) in Wave 2 surveys were analyzed. Table 1 presents participants’ demographic data, the rates of suicidal ideation, and perceived unfavorable attitudes toward homosexuality and same-sex marriage among participants in the surveys. The results of Chi-square test demonstrated that the rates of suicidal ideation among non-heterosexual participants increased from 15.4% in the Wave 1 to 24.4% in the Wave 2 surveys (*p* < 0.001), whereas no significant change was noted in the rates of suicidal ideation among heterosexual participants between the Wave 1 (4.9%) and Wave 2 surveys (6.1%). The ratings of perceived unfavorable attitudes toward homosexuality and same-sex marriage from Taiwanese society, heterosexual friends, and family members significantly increased from the Wave 1 to the Wave 2 surveys in both heterosexual and non-heterosexual participants, except for perceived unfavorable attitudes toward homosexuality from family members. Moreover, 85.9% and 87.0% of heterosexual participants rated themselves as having a favorable attitude toward homosexuality in the Wave 1 and 2 surveys, respectively, indicating that most heterosexual participants in the study were LGB-friendly.

### 3.2. Association of Perceived Unfavorable Attitudes toward Homosexuality and Same-Sex Marriage with Suicidal Ideation in Non-Heterosexual Participants

Table 2 presents the results of the logistic regression analysis of the association of perceived unfavorable attitudes toward homosexuality and same-sexual marriage from Taiwanese society, heterosexual friends, and family members with suicidal ideation in the Wave 1 and 2 surveys among non-heterosexual participants. The results indicate that after the effects of gender and age were controlled, perceived unfavorable attitudes toward homosexuality from these sources were positively associated with suicidal ideation in the Wave 1 (Model I) but not the Wave 2 survey (Model III).

Non-heterosexual participants who perceived an unfavorable attitude toward same-sex marriage among family members were more likely to experience suicidal ideation than those individuals who perceived a favorable attitude among family members in the Wave 1 survey (Model II); furthermore, non-heterosexual participants who perceived an unfavorable attitude toward same-sex marriage from heterosexual friends were more likely to experience suicidal ideation than those individuals who perceived a favorable attitude in these friends in the Wave 2 survey (Model IV).

### 3.3. Association of Perceived Unfavorable Attitudes toward Homosexuality and Same-Sex Marriage with Suicidal Ideation in Heterosexual Participants

Table 3 presents the logistic regression analysis results for the association of perceived unfavorable attitudes toward homosexuality and same-sexual marriage from Taiwanese society, heterosexual friends, and family members with suicidal ideation among heterosexual participants in the surveys. The results indicate that after the effects of gender and age were controlled, heterosexual participants who perceived unfavorable attitudes toward homosexuality from heterosexual friends were more likely to experience suicidal ideation than individuals who perceived favorable attitudes toward homosexuality from these friends in both the Wave 1 (Model V) and Wave 2 surveys (Model VII). These participants who perceived an unfavorable attitude toward same-sex marriage in heterosexual friends were more likely to experience suicidal ideation than those individuals who perceived a favorable attitude toward same-sex marriage in heterosexual friends in both Wave 1 (Model VI) and Wave 2 surveys (Model VIII).

We further divided heterosexual participants into those individuals with favorable (Table 4) and unfavorable (Table 5) attitudes toward homosexuality for analysis. Among heterosexual participants who reported a favorable attitude toward homosexuality (but not those individuals with an unfavorable attitude toward homosexuality), the results revealed a significant association of perceived unfavorable attitudes toward homosexuality and same-sex marriage in their heterosexual friends with suicidal ideation. Perceived unfavorable attitudes toward homosexuality or same-sex marriage from Taiwanese society or family members were not significantly associated with suicidal ideation in heterosexual participants in either survey.

## 4. Discussion

The study results indicate that perceived unfavorable attitudes toward homosexuality from Taiwanese society, heterosexual friends, and family members were positively associated with suicidal ideation among non-heterosexual individuals prior to but not after the same-sex marriage referendums. Among non-heterosexual individuals, perceived unfavorable attitudes toward same-sex marriage in family members before the referendums and in heterosexual friends after the referendums were positively associated with suicidal ideation. Both before and after the referendums, perceived unfavorable attitudes toward homosexuality and same-sex marriage in heterosexual friends were associated with suicidal ideation in heterosexual participants with favorable attitudes but not in those individuals with unfavorable attitudes toward homosexuality. Perceived unfavorable attitudes toward homosexuality and same-sex marriage from Taiwanese society or family members were not significantly associated with suicidal ideation in heterosexual participants.

### 4.1. Unfavorable Attitudes and Suicidal Ideation in Non-Heterosexual Participants

The study revealed that associations of perceived socially unfavorable attitudes and suicidal ideation in the non-heterosexual participants varied by content, source, and survey. Perceived unfavorable attitudes toward homosexuality from Taiwanese society, heterosexual friends, and family members were significantly associated with suicidal ideation prior to the referendums. Although the cross-sectional study design limits the possibility of determining causal relationships, according to ecological systems theory [12], suicidal ideation is influenced by the complex interactions of sexual minority individuals with their environments. Social prejudice and discrimination based on sexual orientation aggravate the suicidal risk of LGB individuals [24]. Perceived unfavorable attitudes toward homosexuality from all of society, heterosexual friends, and family indicate that non-heterosexual individuals are surrounded by negative attitudes toward their sexual orientation. Living in such a comprehensively devaluating environment, they can experience chronic stress and internalize stigmatization, both of which further compromise their mental health and increase the risk of suicidal ideation [8,25]. Non-heterosexual individuals with poor mental health status are especially vulnerable to socially unfavorable attitudes toward their sexual orientation.

The association of perceived unfavorable attitudes toward homosexuality with suicidal ideation became nonsignificant after the same-sex marriage referendums. Instead, perceived unfavorable attitudes toward same-sex marriage were significantly associated with suicidal ideation in non-heterosexual individuals, but the significant source of such attitudes changed from family members before the referendums to heterosexual friends after them. People in Taiwan traditionally require their offspring to continue the family bloodline, and therefore regard same-sex marriage as a challenge to family obligations. Perceived unfavorable attitudes toward same-sex marriage among family members reminds non-heterosexual individuals that they fall short of their families’ expectations, which make them feel burdensome. However, after the referendums, the impact of such attitudes toward same-sex marriage from heterosexual friends seemed to exceed the impact of such attitudes of family members. Groups opposing same-sex marriage in Taiwan spread information against same-sex marriage. The government had an obligation under the Referendum Act to detail its stance prior to the referendums, but the administration kept delaying explaining its position and absented itself from referendum debates. Therefore, Taiwanese people’s understanding of same-sex marriage was misled by the groups opposing it. In social identity threat theories [26], perceived unfavorable attitudes toward same-sex marriage in heterosexual friends can cue a threat to one’s social identity and, in turn, create involuntary stress responses that can compromise one’s mental health.

### 4.2. Unfavorable Attitudes and Suicidal Ideation in Heterosexual Participants

This study demonstrated that perceived unfavorable attitudes toward homosexuality and same-sex marriage from heterosexual friends were associated with suicidal ideation in heterosexual participants both before and after the referendums, but a significant association existed only with those individuals who experience a favorable attitude toward homosexuality. Data from the World Values Survey in Taiwan indicated that since education has improved and liberal values related to marriage, sex, and gender roles have been promoted, overall social tolerance of homosexuality has increased [27]. Heterosexual individuals who are friendly to sexual minorities consider unfavorable attitudes toward homosexuality a social injustice, and their empathy may cause them to feel low. The results suggest that the elimination of discrimination against sexual minorities is important to the mental health of both non-heterosexual and liberal-minded heterosexual individuals.

### 4.3. Implications

Although further study is required to explain the changes in the association of perceived unfavorable attitude toward homosexuality and same-sex marriage with suicidal ideation before and after the referendums, the results of this study indicate that efforts to amend negative attitudes toward homosexuality and same-sex marriage should continue. Currently, in Taiwan, these efforts are mostly made by social movement organizations with human and financial resources that are far more limited than those resources of anti-LGB groups. Given that unfavorable attitudes toward sexual minorities can harm non-heterosexual individuals and lead to the harassment of heterosexual individuals who are LGB-friendly, programs designed to destigmatize sexual minorities should be advanced to the entire population and throughout the country. The government must declare an anti-stigma position with regards to all minorities clearly and definitively and propose effective policies for reducing prejudice against sexual minority individuals.

### 4.4. Limitations

This study has limitations. First, although recruiting participants through Facebook can deliver large numbers of participants quickly, cheaply, and with minimal effort as compared with mail and phone recruitment, access to Facebook is not yet universal, and people are not equally motivated to use it [28]. For example, Facebook has a younger and more progressive audience than the general population. Second, the cross-sectional study design limited the possibility to determine causal relationships between perceived unfavorable attitudes toward homosexuality and same-sex marriage and suicidal ideation. Third, we were unable to determine the mechanism for the change in the pattern of associations between perceived unfavorable attitude toward homosexuality and same-sex marriage and suicidal ideation before and after the same-sex marriage referendums. Fourth, most heterosexual participants in this study reported a favorable attitude toward homosexuality (85.9% in Wave 1 and 87.0% in Wave 2). This is obviously not representative. Further study is warranted to examine associations for attitudes toward homosexuality and same-sex marriage with the mental health of heterosexual individuals.

## 5. Conclusions

Non-heterosexual individuals in Taiwan experience multiple forms of social exclusion regarding their sexual orientation and human rights; in this study, these forms of social exclusion were significantly associated with suicidal ideation. Heterosexual individuals who were friendly to sexual minorities were also vulnerable to peers’ unfavorable attitudes toward homosexuality and same-sex marriage. Programs for destigmatizing sexual minorities are urgently required to address social equality and improve the mental health of affected parties.

## Figures and Tables

**Table 1 ijerph-17-01047-t001:** Gender, age, suicidal ideation, and perceived attitudes toward homosexuality and same-sex marriage.

Variables	Non-Heterosexual			Heterosexual		
	Wave 1 (*N* = 1796)	Wave 2 (*N* = 798)	*p*	Wave 1 (*N* = 1443)	Wave 2 (*N* = 539)	*p*
Gender, *n* (%) ^a^						
Male	879 (48.9)	386 (48.4)	0.788	311 (21.6)	123 (22.8)	0.544
Female	917 (51.1)	412 (51.6)		1132 (78.4)	416 (77.2)	
Age (years), mean (SD) ^b^	29.1 (7.0)	29.5 (7.2)	0.210	32.4 (8.9)	36.3 (10.2)	<0.001
Suicidal ideation, *n* (%) ^a^						
No	1520 (84.6)	603 (75.6)	<0.001	1372 (95.1)	506 (93.9)	0.286
Yes	276 (15.4)	195 (24.4)		71 (4.9)	33 (6.1)	
Taiwanese society’s attitude toward homosexuality, *n* (%) ^a^						
Favorable	698 (38.9)	112 (14.0)	<0.001	472 (32.7)	53 (9.8)	<0.001
Unfavorable	1098 (61.1)	686 (86.0)		971 (67.3)	486 (90.2)	
Heterosexual friends’ attitude toward homosexuality, *n* (%) ^a^						
Favorable	1469 (81.8)	611 (76.6)	0.002	1074 (74.4)	358 (66.4)	<0.001
Unfavorable	327 (18.2)	187 (23.4)		369 (25.6)	181 (33.6)	
Family members’ attitude toward homosexuality, *n* (%) ^a^						
Favorable	641 (35.7)	271 (34.0)	0.394	516 (35.8)	187 (34.7)	0.659
Unfavorable	1155 (64.3)	527 (66.0)		927 (64.2)	352 (65.3)	
Taiwanese society’s attitude toward same-sex marriage, *n* (%) ^a^						
Favorable	687 (38.3)	49 (6.1)	<0.001	420 (29.1)	22 (4.1)	<0.001
Unfavorable	1109 (61.7)	749 (93.9)		1023 (70.9)	517 (95.9)	
Heterosexual friends’ attitude toward same-sex marriage, *n* (%) ^a^						
Favorable	1430 (79.6)	559 (70.1)	<0.001	1003 (69.5)	321 (59.6)	<0.001
Unfavorable	366 (20.4)	239 (29.9)		440 (30.5)	218 (40.4)	
Family members’ attitude toward same-sex marriage, *n* (%) ^a^						
Favorable	670 (37.3)	252 (31.6)	0.005	532 (36.9)	169 (31.4)	0.022
Unfavorable	1126 (62.7)	546 (68.4)		911 (63.1)	370 (68.6)	
Heterosexual participants’ attitude toward homosexuality, *n* (%) ^a^						
Favorable				1239 (85.9)	469 (87.0)	0.509
Unfavorable				204 (14.1)	70 (13.0)	

^a^, chi-square test; ^b^, *t* test.

**Table 2 ijerph-17-01047-t002:** Association of perceived unfavorable attitudes toward homosexuality and same-sex marriage from Taiwanese society, heterosexual friends, and family members with suicidal ideation in non-heterosexual participants.

Variables	Suicidal Ideation											
	Wave 1						Wave 2					
	Model I			Model II			Model III			Model IV		
	Wals χ^2^	OR	95% CI	Wals χ^2^	OR	95% CI	Wals χ^2^	OR	95% CI	Wals χ^2^	OR	95% CI
Gender	2.743	0.800	0.614–1.042	3.157	0.789	0.607–1.025	0.302	1.098	0.787–1.530	0.421	1.116	0.801–1.556
Age	11.252 **	0.966	0.946–0.986	10.889 **	0.966	0.946–0.986	6.932 **	0.967	0.943–.991	8.886 **	0.963	0.939–0.987
*Perceived unfavorable attitude toward homosexuality from*												
Taiwanese society	**4.007 ***	**1.342**	**1.006–1.789**				1.681	1.408	0.839–2.361			
Heterosexual friends	**7.950 ****	**1.579**	**1.149–2.170**				1.063	1.238	0.825–1.856			
Family members	**8.425 ****	**1.563**	**1.156–2.113**				0.189	1.083	0.756–1.552			
*Perceived unfavorable attitude toward same-sex marriage from*												
Taiwanese society				0.313	1.084	0.817–1.439				0.869	1.431	0.674–3.039
Heterosexual friends				3.311	1.349	0.977–1.862				**5.836 ***	**1.588**	**1.091–2.312**
Family members				**4.935 ***	**1.384**	**1.039–1.844**				1.277	0.812	0.565–1.166

CI: confidence interval; OR: odds ratio. ***: *p* < 0.05; **: *p* < 0.01; ***: *p* < 0.001; The bold shows the significant associations between suicidal ideation and perceived unfavorable attitudes toward homosexuality and same-sex marriage in the Wave 1 and Wave 2 surveys.

**Table 3 ijerph-17-01047-t003:** Association of perceived unfavorable attitudes toward homosexuality and same-sex marriage from Taiwanese society, heterosexual friends, and family members with suicidal ideation in heterosexual participants.

	Suicidal Ideation											
	Wave 1						Wave 2					
	Model V			Model VI			Model VII			Model VIII		
	Wals χ^2^	OR	95% CI	Wals χ^2^	OR	95% CI	Wals χ^2^	OR	95% CI	Wals χ^2^	OR	95% CI
Gender	0.001	1.009	0.559–1.823	0.000	0.994	0.551–1.793	0.026	1.080	0.420–2.777	0.006	1.037	0.401–2.680
Age	15.584 ***	0.930	0.898–0.964	15.452 ***	0.930	0.897–0.964	19.578 ***	0.890	0.845–0.937	20.419 ***	0.890	0.846–0.936
*Perceived unfavorable attitude toward homosexuality from*												
Taiwanese society	0.001	1.008	0.586–1.734				0.879	2.673	0.342–20.875			
Heterosexual friends	**5.193 ***	**1.959**	**1.099–3.493**				**9.814 ****	**3.805**	**1.649–8.778**			
Family members	0.006	1.021	0.596–1.748				0.009	1.041	0.450–2.409			
*Perceived unfavorable attitude toward same-sex marriage from*												
Taiwanese society				0.110	0.911	0.525–1.580				0.153	0.656	0.080–5.417
Heterosexual friends				**5.605 ***	**2.021**	**1.129–3.619**				**11.700 ***	**4.204**	**1.846–9.574**
Family members				1.049	0.760	0.449–1.285				0.385	0.771	0.338–1.756

CI, confidence interval; OR, odds ratio; *, *p* < 0.05; **, *p* < 0.01; ***, *p* < 0.001; The bold shows the significant associations between suicidal ideation and perceived unfavorable attitudes toward homosexuality and same-sex marriage in the Wave 1 and Wave 2 surveys.

**Table 4 ijerph-17-01047-t004:** Association of perceived unfavorable attitudes toward homosexuality and same-sex marriage from Taiwanese society, heterosexual friends, and family members with suicidal ideation in heterosexual participants with a favorable attitude toward homosexuality.

	Suicidal ideation											
	Wave 1						Wave 2					
	Model IX			Model X			Model XI			Model XII		
	Wals χ^2^	OR	95% CI	Wals χ^2^	OR	95% CI	Wals χ^2^	OR	95% CI	Wals χ^2^	OR	95% CI
Gender	0.491	1.301	0.624–2.713	0.000	0.994	0.551–1.793	0.041	1.112	0.399–3.098	0.051	1.128	0.397–3.206
Age	22.637 ***	0.892	0.851–.935	15.452 ***	0.930	0.897–0.964	19.215 ***	0.875	0.824–0.929	20.556 ***	0.873	0.823–0.926
*Perceived unfavorable attitude toward homosexuality from*												
Taiwanese society	0.004	0.982	0.551–1.749				0.492	2.103	0.263–16.781			
Heterosexual friends	**4.664 ***	**2.179**	**1.075–4.419**				**11.430 ****	**4.654**	**1.908–11.349**			
Family members	0.031	1.052	0.596–1.856				0.091	1.140	0.487–2.669			
*Perceived unfavorable attitude toward same-sex marriage from*												
Taiwanese society				1.043	0.577–1.886	1.043				0.338	0.529	0.062–4.528
Heterosexual friends				**2.681 ***	**1.363–5.273**	**2.681**				**13.929 *****	**5.189**	**2.186–12.319**
Family members				0.581	0.332–1.017	0.581				0.194	0.829	0.360–1.909

CI, confidence interval; OR, odds ratio; *, *p* < 0.05; **, *p* < 0.01; ***, *p* < 0.001; The bold shows the significant associations between suicidal ideation and perceived unfavorable attitudes toward homosexuality and same-sex marriage in the Wave 1 and Wave 2 surveys.

**Table 5 ijerph-17-01047-t005:** Association of perceived unfavorable attitudes toward homosexuality and same-sex marriage from Taiwanese society, heterosexual friends, and family members with suicidal ideation in heterosexual participants with an unfavorable attitude toward homosexuality.

	Suicidal ideation											
	Wave 1						Wave 2					
	Model XIII			Model XIV			Model XV			Model XVI		
	Wals χ^2^	OR	95% CI	Wals χ^2^	OR	95% CI	Wals χ^2^	OR	95% CI	Wals χ^2^	OR	95% CI
Gender	1.062	0.547	0.173–1.724	0.657	0.616	0.191h1.987	0.046	0.716	0.034–14.887	0.086	0.626	0.027–14.365
Age	0.244	1.014	0.958–1.074	0.495	1.021	0.963–1.084	0.149	0.973	0.848–1.117	0.034	0.986	0.851–1.143
*Perceived unfavorable attitude toward homosexuality from*												
Taiwanese society	0.113	1.381	0.210–9.058				0.000 ^a^	–	–			
Heterosexual friends	0.119	1.341	0.253–7.110				0.000 ^a^	–	–			
Family members	1.637	0.278	0.039–1.975				0.000 ^a^	–	–			
*Perceived unfavorable attitude toward same-sex marriage from*												
Taiwanese society				1.059	0.417	0.079–2.204				0.000 ^a^	-	-
Heterosexual friends				0.705	0.533	0.123–2.312				0.000 ^a^	-	-
Family members				0.000 ^a^	-	-				0.000 ^a^	-	-

CI, confidence interval; OR, odds ratio; ^a^, no participant perceiving social favorable attitudes toward homosexuality or same-sex marriage and having suicidal ideation.

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
