# Peer review of "Associations of Perceived Socially Unfavorable Attitudes toward Homosexuality and Same-Sex Marriage with Suicidal Ideation in Taiwanese People before and after Same-Sex Marriage Referendums"

_ijerph, 2020, doi:10.3390/ijerph17031047_

Round 1
Reviewer 1 Report
I am reviewing the paper entitled “Associations of Perceived Socially Unfavorable Attitudes Toward Homosexuality and Same-Sex Marriage with Suicidal Ideation in Taiwanese People before and after Same-Sex Marriage Referendums” submitted to International Journal of Environmental Research and Public Health is a solid paper with interesting policy implications. Although the papers is very well written, I make a few suggestions throughout the paper to improve the prose.
Lines 38 and 39, the authors should say “non-heterosexual”. Line 44 should say “those individuals”. Line 55 should say “report significantly higher”. Line 78 should say “attitudes toward same-sex marriage are associated”. Line 80 should say “various sources show different”. Line 85 should say “individuals experience significant”. Line 90 should say “who report a friendly attitude”. Line 91 should say “justice and, therefore, they may be upset by them.” Line 112 says “spent much more funds”. This phrasing is strange, but I do not have any suggestions.
Line 120 should say “in Taiwan, which was a two-wave”. Line 125 and 126 should say “We tested four hypotheses.” Line 128 should say “members would be positively associated”. Line 131 should say “attitudes would influence suicidal ideation” and it should say “non-heterosexual”. Line 132 should say “heterosexual individuals may show differing”. Line 133 should say “ideation would vary”. Line 135 should say “ideation would be different”.
Line 144 should say “effective recruitment metrics”. Line 155 should say “The IRB, thus, agreed”. Line 163 should say “classified as experiencing significant suicidal ideation”. Line 186 should say “using a descriptive analysis”. Line 188 should say “non-heterosexual”. Line 195 should say “The data for 3,239 participants”. Line 207 should say “second survey.” Line 212 should say “members were”. Line 213 should say “second survey”. Line 227 should say “Non-heterosexual”. Line 228 should say “likely to experience suicidal ideation than those individuals who”. Line 229 should say “favorable attitude among family members…non-heterosexual”. Line 231 should say “likely to experience suicidal ideation than those individuals”.
Line 243 should say “attitudes toward”. Line 244 should say “likely to experience suicidal ideation than individuals who perceived favorable attitudes”. Line 247 should say “likely to experience suicidal ideation than those individuals”. Line 250 should say “those individuals”. Line 251 should say “participants who report a favorable”. Line 252 should say “but not those individuals with an unfavorable attitude”. Lines 267 and 268 should say “non-heterosexual”. Line 273 should say “those individuals with”. Line 289 should say “Non-heterosexual”. Line 296 should say “and, therefore, regard”. Line 297 should say “unfavorable attitudes toward”. Line 298 should say “non-heterosexual”. Line 299 should say “expectations, which make them feel burdensome”. Line 300 should say “exceed the impact of such attitudes of family members.” Line 306 should say “attitudes toward”. Line 312 should say “those individuals who experience a favorable”. Line 313 should say “Taiwan indicated that”. Line 318 should say “non-heterosexual and liberal-minded heterosexual individuals.” Line 325 should say “limited than those resources”. Line 326 should say “minorities may harm non-heterosexual individuals and lead”. Lie 345 should say “Non-heterosexual”. Line 346 should say “these forms of social exclusion were significantly associated”.
Author Response
We appreciate your comments on our manuscript. As discussed below, we have revised our manuscript with underlines according to the reviewers. The following responses have been prepared to address your comments in a point-by-point fashion. Please let us know if there is anything else we should provide.
Comment
Lines 38 and 39, the authors should say “non-heterosexual”.
Response
We replaced “nonheterosexual” by “non-heterosexual” thorough the revised manuscript.
Comment
Line 44 should say “those individuals”.
Response
We revised it. Please refer to line 43.
Comment
Line 55 should say “report significantly higher”.
Response
We revised it. Please refer to line 54.
Comment
Line 78 should say “attitudes toward same-sex marriage are associated”.
Response
We revised it. Please refer to line 75.
Comment
Line 80 should say “various sources show different”.
Response
We revised it. Please refer to line 77.
Comment
Line 85 should say “individuals experience significant”.
Response
We revised it. Please refer to line 82-83.
Comment
Line 90 should say “who report a friendly attitude”.
Response
We revised it. Please refer to line 87.
Comment
Line 91 should say “justice and, therefore, they may be upset by them.”
Response
We revised it. Please refer to line 88-89.
Comment
Line 112 says “spent much more funds”. This phrasing is strange, but I do not have any suggestions.
Response
We deleted this sentence in the revised manuscript. Please refer to line 107.
Comment
Line 120 should say “in Taiwan, which was a two-wave”.
Response
We revised it. Please refer to line 113.
Comment
Line 125 and 126 should say “We tested four hypotheses.”
Response
We revised it. Please refer to line 118-119.
Comment
Line 128 should say “members would be positively associated”.
Response
We revised it. Please refer to line 121.
Comment
Line 131 should say “attitudes would influence suicidal ideation” and it should say “non-heterosexual”.
Response
We revised it. Please refer to line 124.
Comment
Line 132 should say “heterosexual individuals may show differing”.
Response
We revised it. Please refer to line 125.
Comment
Line 133 should say “ideation would vary”.
Response
We revised it. Please refer to line 126-127.
Comment
Line 135 should say “ideation would be different”.
Response
We revised it. Please refer to line 129.
Comment
Line 144 should say “effective recruitment metrics”.
Response
We revised it. Please refer to line 137.
Comment
Line 155 should say “The IRB, thus, agreed”.
Response
We revised it. Please refer to line 148.
Comment
Line 163 should say “classified as experiencing significant suicidal ideation”.
Response
We revised it. Please refer to line 156.
Comment
Line 186 should say “using a descriptive analysis”.
Response
We revised it into “using chi-square and t tests”. Please refer to line 180.
Comment
Line 188 should say “non-heterosexual”.
Response
We replaced “nonheterosexual” by “non-heterosexual” thorough the revised manuscript.
Comment
Line 195 should say “The data for 3,239 participants”.
Response
We revised it. Please refer to line 189.
Comment
Line 207 should say “second survey.”
Response
We revised this paragraph. Please refer to line 193-200.
Comment
Line 212 should say “members were”.
Response
We revised this paragraph. Please refer to line 193-200.
Comment
Line 213 should say “second survey”.
Response
We revised this paragraph. Please refer to line 193-200.
Comment
Line 227 should say “Non-heterosexual”.
Response
We replaced “nonheterosexual” by “non-heterosexual” thorough the revised manuscript.
Comment
Line 228 should say “likely to experience suicidal ideation than those individuals who”.
Response
We revised it. Please refer to line 217.
Comment
Line 229 should say “favorable attitude among family members…non-heterosexual”.
Response
We revised it. Please refer to line 218.
Comment
Line 231 should say “likely to experience suicidal ideation than those individuals”.
Response
We revised it. Please refer to line 220-221.
Comment
Line 243 should say “attitudes toward”.
Response
We revised this paragraph. Please refer to line 233.
Comment
Line 244 should say “likely to experience suicidal ideation than individuals who perceived favorable attitudes”.
Response
We revised this paragraph. Please refer to line 234-235.
Comment
Line 247 should say “likely to experience suicidal ideation than those individuals”.
Response
We revised this paragraph. Please refer to line 237-238.
Comment
Line 250 should say “those individuals”.
Response
We revised this paragraph. Please refer to line 240.
Comment
Line 251 should say “participants who report a favorable”.
Response
We revised this paragraph. Please refer to line 242.
Comment
Line 252 should say “but not those individuals with an unfavorable attitude”.
Response
We revised this paragraph. Please refer to line 242-243.
Comment
Lines 267 and 268 should say “non-heterosexual”.
Response
We replaced “nonheterosexual” by “non-heterosexual” thorough the revised manuscript.
Comment
Line 273 should say “those individuals with”.
Response
We revised this paragraph. Please refer to line 271.
Comment
Line 289 should say “Non-heterosexual”.
Response
We replaced “nonheterosexual” by “non-heterosexual” thorough the revised manuscript.
Comment
Line 296 should say “and, therefore, regard”.
Response
We revised this paragraph. Please refer to line 296.
Comment
Line 297 should say “unfavorable attitudes toward”.
Response
We revised this paragraph. Please refer to line 297.
Comment
Line 298 should say “non-heterosexual”.
Response
We replaced “nonheterosexual” by “non-heterosexual” thorough the revised manuscript.
Comment
Line 299 should say “expectations, which make them feel burdensome”.
Response
We revised this paragraph. Please refer to line 299.
Comment
Line 300 should say “exceed the impact of such attitudes of family members.”
Response
We revised this paragraph. Please refer to line 300-301.
Comment
Line 306 should say “attitudes toward”.
Response
We revised this paragraph. Please refer to line 306.
Comment
Line 312 should say “those individuals who experience a favorable”.
Response
We revised this paragraph. Please refer to line 313.
Comment
Line 313 should say “Taiwan indicated that”.
Response
We revised this paragraph. Please refer to line 314.
Comment
Line 318 should say “non-heterosexual and liberal-minded heterosexual individuals.”
Response
We revised this paragraph. Please refer to line 319-320.
Comment
Line 325 should say “limited than those resources”.
Response
We revised this paragraph. Please refer to line 327.
Comment
Line 326 should say “minorities may harm non-heterosexual individuals and lead”.
Response
We revised this paragraph. Please refer to line 328.
Comment
Lie 345 should say “Non-heterosexual”.
Response
We replaced “nonheterosexual” by “non-heterosexual” thorough the revised manuscript.
Comment
Line 346 should say “these forms of social exclusion were significantly associated”.
Response
We revised this paragraph. Please refer to line 350-351.
Reviewer 2 Report
In this empirical report, the authors surveyed adults regarding their perceptions of societal, peer, and familial attitudes toward homosexuality and same-sex marriage, before (wave 1) and shortly after (wave 2) a set of highly visible and politically charged same-sex marriage referenda in Taiwan. Participants were recruited from Facebook advertisements and were generally in their late 20s or early 30s.
This study addresses an important topic and has some interesting findings. I have some questions/suggestions for improvement.
In the introduction, in section 1.2, the sentence beginning “same-sex marriage ban are advocated” (lines 76-8) reads more like political commentary than scientific reasoning and the authors may consider excluding it. Likewise, all of section 1.4 could be shortened to a brief summary of the facts and timeline from 2017 to the referenda. Para 2 (lines 107-17) also sounds at times like political commentary and reduce the objectivity of the study. In the methods, were there any safeguards to protect respondents who endorsed suicidal ideation, such as instructions to seek immediate help and a suicide hotline number? Why were attitudes dichotomized into unfavorable and favorable? Wouldn’t a continuous scale allow greater statistical power? Linear instead of logistic regression models could be used. Why were heterosexual respondents only asked about their attitudes toward homosexuality and not about same-sex marriage? Was there an option for respondents to identify with a non-binary gender? In the results, section 3.1 is redundant with table 1; the text should highlight the most significant or notable findings. Likewise, the authors should consider showing chi-squared values for each variable in wave 1 and 2 in table 1 so as to highlight any significant changes in distribution between waves 1 and 2. For section 3.3, the authors should consider providing 2 supplementary tables with table 3 values for heterosexual respondents with unfavorable and favorable attitudes toward homosexuality. In section 4.1, lines 301-7 again sound more like political commentary than scientific reasoning and reduce the objectivity of this report. In section 4.4, the authors might briefly surmise how selection bias may have influenced the results (eg, Facebook may have a younger and more progressive audience than the general population).
Author Response
Reviewer 2
We appreciate your comments on our manuscript. As discussed below, we have revised our manuscript with underlines according to the reviewers. The following responses have been prepared to address your comments in a point-by-point fashion. Please let us know if there is anything else we should provide.
Comment
In the introduction, in section 1.2, the sentence beginning “same-sex marriage ban are advocated” (lines 76-8) reads more like political commentary than scientific reasoning and the authors may consider excluding it.
Response
Thank you for your suggestion. We excluded this sentence from the revised manuscript. Please refer to line 75.
Comment
Likewise, all of section 1.4 could be shortened to a brief summary of the facts and timeline from 2017 to the referenda. Para 2 (lines 107-17) also sounds at times like political commentary and reduce the objectivity of the study.
Response
Thank you for your suggestion. We shortened the content of this section and deleted some sentences to keep the objectivity of the study. Please refer to line 107.
Comment
In the methods, were there any safeguards to protect respondents who endorsed suicidal ideation, such as instructions to seek immediate help and a suicide hotline number?
Response
Thank you for your reminding. This is the first time for us to do online surveys and we did not include any instruction for participants who endorsed suicidal ideation to seek immediate help. We will keep your suggestion in mind and include instructions to seek immediate help in next online surveys.
Comment
Why were attitudes dichotomized into unfavorable and favorable? Wouldn’t a continuous scale allow greater statistical power? Linear instead of logistic regression models could be used.
Response
The responses to the six questions for social attitudes toward homosexuality and same-sex marriage were significantly skewed. Therefore, we dichotomized the responses to the questions into unfavorable and favorable attitudes for further analysis. We added the explanation into the revised manuscript. Please refer to line 166-167.
Comment
Why were heterosexual respondents only asked about their attitudes toward homosexuality and not about same-sex marriage?
Response
We have asked the participants about their attitudes toward same-sex marriage and have examined the relationship between mental health and attitude toward same-sex marriage in the first survey (Yu-Te Huang, Mu-Hong Chen, Huei-Fan Hu, Nai-Ying Ko, Cheng-Fang Yen. Role of among people in Taiwan. Journal of the Formosan Medical Association 2020;119:150-156). The attitudes toward homosexuality and toward same-sex marriage are highly correlated to each other. The present study used attitudes toward homosexuality only for analysis.
Comment
Was there an option for respondents to identify with a non-binary gender?
Response
In the questionnaire the responses for the question for gender include male, female and transgender. However, only 1.4% and 2.4% of participants were transgender, respectively. The small number limited the possibility of statistical analysis. Therefore, we selected male and female participants for analysis in the present study.
Comment
In the results, section 3.1 is redundant with table 1; the text should highlight the most significant or notable findings. Likewise, the authors should consider showing chi-squared values for each variable in wave 1 and 2 in table 1 so as to highlight any significant changes in distribution between waves 1 and 2.
Response
Thank you for your suggestion. In the revised manuscript we compared gender, age, suicidal ideation, and perceived unfavorable attitudes toward homosexuality and same-sex marriage between the Wave 1 and Wave 2 surveys using chi-square and t tests. We revised the contents of section 3.1. to highlight the significant changes of suicidal ideation and perceived unfavorable attitudes toward homosexuality and same-sex marriage between the Wave 1 and Wave 2 surveys. Please refer to line 193-200 and Table 1.
Comment
For section 3.3, the authors should consider providing 2 supplementary tables with table 3 values for heterosexual respondents with unfavorable and favorable attitudes toward homosexuality.
Response
We added Table 4 and Table 5 to describe the values for heterosexual respondents with unfavorable and favorable attitudes toward homosexuality. Please refer to line 240-243, Table 4 and Table 5.
Comment
In section 4.1, lines 301-7 again sound more like political commentary than scientific reasoning and reduce the objectivity of this report.
Response
We revised the sentence into “Groups opposing same-sex marriage in Taiwan spread information against same-sex marriage.” Please refer to line 301-302.
Comment
In section 4.4, the authors might briefly surmise how selection bias may have influenced the results (eg, Facebook may have a younger and more progressive audience than the general population).
Response
Thank you for your suggestion. We added the description into the revised manuscript. Please refer to line 338-339.
Round 2
Reviewer 2 Report
This version addresses my suggestions.